# The Role of Adult Vaccines as Part of Antimicrobial Stewardship: A Scoping Review

**DOI:** 10.3390/antibiotics12091429

**Published:** 2023-09-10

**Authors:** Charles Travers Williams, Syed Tabish Razi Zaidi, Bandana Saini, Ronald Castelino

**Affiliations:** 1Faculty of Medicine and Health, University of Sydney, Camperdown 2050, Australiaronald.castelino@sydney.edu.au (R.C.); 2Faculty of Medicine and Health, University of Leeds, Leeds LS2 9JT, UK

**Keywords:** antimicrobial stewardship, vaccines, antimicrobial resistance, scoping review

## Abstract

Background: Antimicrobial resistance (AMR) is a significant global health concern, causing an estimated 700,000 deaths annually. Although immunisation has been shown to significantly reduce AMR, the role of vaccines as part of antimicrobial stewardship (AMS) practices is often overlooked. Objective: To identify and examine the available literature on the role of vaccines as part of AMS practices. Method: A scoping review was conducted in the following databases: MEDLINE, Embase, Scopus, CINAHL, CCRCT, IPA, and WoS, along with grey literature sources. The review was conducted using the JBI Methodology for Scoping Reviews and reported in line with the PRISMA-SCr checklist. Results: Among the 1711 records identified, 34 met the inclusion criteria; 8 discussed only the concept, while 26 discussed both the concept and the vaccine implementation method in AMS practices. There were eight recommended and/or utilised types of AMS activities identified involving vaccines, under four key themes of vaccine-related AMS strategies: Education, Screening, Vaccination, and Monitoring. Influenza and pneumococcal vaccines had the most evidence for inclusion. Conclusion: Overall, the evidence supports the role of vaccines as part of AMS practices and the value of their inclusion in creating improved and comprehensive AMS strategies to further combat the development of AMR.

## 1. Introduction

Antimicrobial resistance (AMR) is defined as the ability of infection-causing micro-organisms to adapt and survive despite adequate antimicrobial treatment [1]. Stemming from the overuse and misuse of antimicrobials, AMR remains a major global health concern and is estimated to be responsible for 700,000 deaths per year worldwide. Without intervention, by 2050, it is projected that AMR-consequent deaths will increase to 10 million per year globally [1].

Vaccines are widely recognised as one of the most successful and cost-effective healthcare interventions, saving an estimated four million lives globally each year [2]. Often thought of solely as an infection prevention and control measure, vaccination as part of antimicrobial stewardship (AMS) strategies are often overlooked. However, data show that vaccines can reduce AMR by reducing both the incidence of bacterial infections and the over and/or inappropriate use of antimicrobials [3]. For example, a large USA study analysed data recorded from outpatient/ambulatory hospital patient visits and reported that 50% of the antibiotic prescriptions were inappropriately given for respiratory illnesses associated with viral pathogens, such as influenza [4]. On the other hand, evidence from a systematic review highlighted that influenza vaccination can reduce antibiotic use among healthy adults by nearly one-third [5]. Similarly, the introduction of the *Haemophilus influenzae* type b (Hib) and 13-valent pneumococcal conjugate (PCV13) vaccines have also been shown to reduce both antibiotic use and the prevalence of resistant strains [3,5,6,7].

However, despite saving millions of lives each year and reducing the inappropriate use of antibiotics, AMR, and healthcare costs, vaccination uptake remains low among adults and far below national targets, thus posing a significant public health concern [8,9]. For instance, the recent pandemic was subject to much heated debate in both public and social media domains in many countries, leading to varied uptake of the effective range of COVID-19 vaccines globally. Whilst several international guidelines list vaccination programs as a key element of AMS toolkits, the intense focus of most AMS programs remains on antimicrobial use rather than prevention strategies [10]. As such, the role of vaccines as part of AMS strategies presents an opportunity for healthcare organisations and warrants further investigation.

To the best of our knowledge, there are no systematic or scoping reviews available on the use of vaccines as a component of AMS strategies. Considering the potential positive impact of vaccines on AMR, healthcare costs, and overall patient health outcomes, there is a need for a comprehensive scoping review to gather and analyse the available literature on this concept. Therefore, the aim of this scoping review is to identify and examine the published evidence pertaining to the role of vaccines as part of AMS practices, with the view to characterise the existing knowledge in this field.

## 2. Materials and Methods

This scoping review was conducted in accordance with the Joanna Briggs Institute (JBI) Methodology for Scoping Reviews [11]. The methodological approach was selected based on the scarcity of information on the concept in question and the need to broadly examine, identify, and map the nature of the available evidence on vaccines as part of AMS practices. The review is reported in line with the Preferred Reporting Items for Systematic Review and Meta-Analysis Extension for Scoping Reviews (PRISMA-ScR) checklist [12]. The protocol (see Appendix A) was registered on the Open Science Framework, under the DOI: osf.io/7nh42, on 11 September 2021.

### 2.1. Eligibility Criteria and Information Sources

A comprehensive literature search was performed in seven databases (MEDLINE, Embase, Scopus, CINAHL, Cochrane Central Register of Controlled Trials [CCRCT], International Pharmaceutical Abstracts [IPA], and Web of Science [WoS]) and limited grey literature (.org, .edu, .gov, .int, .au websites) for information that describes the use of vaccines in adults (≥18 years of age) to reduce AMR as part of AMS strategies. The scoping review considered the concept of vaccines as part of AMS in adult individuals or patients and the strategies used to support implementation and uptake. In addition, barriers and gaps to vaccination as part of AMS interventions were also examined (research questions are outlined in Appendix A).

All sources of evidence, including primary studies, systematic reviews, reports, dissertations, and expert opinion articles published in the English language were considered for inclusion without restriction on the year of publication or geographical locations. Literature discussing the use of vaccines in adolescent or paediatric populations (<18 years of age) were excluded, as more robust immunisation schedules are often in place and certain vaccinations are mandated in this population group (see Table 1). Furthermore, children and adolescents are often not the primary decision-maker when it comes to vaccinations.

### 2.2. Search Strategy

The search strategy was developed and refined with the assistance of a research librarian. The search strategy consisted of two themes: vaccines and AMS. A preliminary search was initially performed in MEDLINE and Embase, and additional relevant free-text terms and subject headings were collected to revise the search strategy accordingly. The final search was conducted on 23 April 2021 and included all relevant literature up to that date. The complete details of the search strategy are available in the Appendix A.

### 2.3. Selection of Sources of Evidence, Data Charting Process, and Data Items

Records retrieved from the electronic databases were exported into EndNote X9 and duplications were removed. A 10% sample of the records were screened to validate the search strategy before proceeding to full screening by two independent reviewers (RC and TZ). A two-step screening process was implemented to assess the literature for relevance and inclusion. The title and abstract were first screened, and subsequently included, when found to be relevant to the research question. Once completed, full-text article screening was performed and assessed for relevance to the research question. Any disagreements for inclusion between the independent reviewers were resolved through discussion and consensus. Any discrepancies that could not be resolved through discussion were resolved by involving a third independent reviewer, who made the final call.

To chart the data from the included records, information on the author, year of publication, country of origin, type of study/study design, aim/purpose, population, key findings/recommendations, and gaps/limitations were extracted.

### 2.4. Critical Appraisal of Individual Sources of Evidence

Risk of bias appraisal was not performed on the included studies because this was not the aim of this scoping review.

### 2.5. Synthesis of Results

Given the heterogeneity of the identified records, descriptive statistics were used to summarise the characteristics and the results presented in a combination of both tables and graphical charts. A narrative summary of the available literature on the role of vaccines as part of AMS practices is presented.

## 3. Results

### 3.1. Study Selection

A total of 1711 records were identified through the search strategy performed on the seven main databases. Additionally, 91 records were identified through the grey literature searches and screening of the reference lists of the included studies. After the removal of duplicates, 1130 records were considered for initial title and abstract screening. Following full-text screening, 34 research articles were eligible for inclusion in the review (Figure 1). The identified articles were further divided into two groups: articles that discussed only the concept (n = 8) and articles that discussed both the concept and vaccine-related AMS interventions (n = 26).

### 3.2. Study Characteristics

#### 3.2.1. Articles That Discussed Only the Concept of Vaccines as Part of AMS Practices

All eight articles that discussed only the concept of vaccines as part of AMS practices were published within the last decade, with the earliest report being published in 2013 (Table 2). The articles were from eight different countries (Argentina, Germany, Iran, Singapore, South Africa, Switzerland, the UK, and the USA), with the earliest publication being from the UK. The evidence type was primarily expert opinion (n = 5), followed by prospective (n = 1) and retrospective observational studies (n = 1), respectively. Additionally, one guideline discussing the concept of vaccination as an AMS strategy was also identified. Among the eight articles, most were general and conducted across healthcare settings (n = 3). The other studies were conducted in both a hospital and community setting (n = 2), in a hospital alone (n = 2), or in a community setting alone (n = 1). One study was specifically in the context of gut microbiome support, and another in the context of acute exacerbations of chronic obstructive pulmonary disorder (AECOPD).

#### 3.2.2. Articles Discussing Vaccines as Part of AMS Practices

A total of 26 articles discussed how vaccines could be included as part of AMS practices, and all were published within the last decade (2012–2021; Table 3). Most of the articles were conducted in the USA (n = 12) and Australia (n = 4). The study types included expert opinions (n = 9), guidelines and recommendations (n = 5), systematic reviews (n = 4), interventional studies (n = 2), and a cross-sectional survey (n = 1), which discussed the concept with methods of implementation of vaccinations as an AMS strategy. There were also various editorials (n = 5); however, they were all non-peer reviewed. Most of the studies were conducted across healthcare settings (n = 9), in a community setting alone (n = 7), or in both hospital and community settings (n = 4). Other settings included hospital alone (n = 3), aged care alone (n = 1), community and aged care (n = 1), and the general public (n = 1). One study was specifically in the context of upper respiratory tract and urinary tract infections, and another in community-acquired pneumonia. Most of the studies focused on the role of pharmacists (n = 8) and nurses (n = 3) as antimicrobial stewards.

### 3.3. Key Findings

#### 3.3.1. Types of Vaccines Discussed and Key Evidence to Support Inclusion into AMS Practices

Across all articles, the two main vaccines discussed for inclusion into AMS practices were influenza (n = 25) and pneumococcal vaccines (n = 19; Figure 2). Darazam et al. stated in a narrative review that there was definitive evidence to consider annual influenza vaccination as part of AMS programs [15], and referenced data that showed a 64% reduction in influenza-associated antibiotic prescriptions with the introduction of the universal flu immunisation program in Ontario, Canada [47]. Smith et al. estimated that influenza vaccination prevented 10.6% and avoided 7.3% of acute respiratory tract illness and antibiotic prescriptions, respectively [18]. A systematic review by Doherty et al. reported a 52% unadjusted median reduction in antimicrobial use with influenza vaccination in adults [27]. Similarly, an expert opinion by Brink AJ [24] highlighted evidence which demonstrated that the post-partum influenza vaccination of mothers was associated with a 45.4% reduction in antibiotic prescriptions in children, along with a reduction in acute respiratory illness, febrile episodes, and influenza-like episodes [48]. Finally, a systematic review by the Pharmacy Accountability Measures (PAM) Work Group identified influenza vaccination as a key measure and was endorsed by the National Quality Forum (NQF) for pharmacy across all healthcare settings [23]. Influenza vaccines were selected due to the morbidity and mortality associated with the disease, and the avoidance of unnecessary antibiotic treatment in adults for acute bronchitis and other respiratory infections. NQF-endorsed measures were alleged to save the USA healthcare system almost USD 30 billion each year [23].

Regarding pneumococcal vaccines, a narrative review by O’Brien et al. emphasised that the pneumococcal immunisations not only reduced resistant *Streptococcus pneumoniae* in children, but also in the general population, because of reduced infection transmission [17]. Brink et al. highlighted that one of the best examples supporting the role of vaccines to reduce AMR and the need for inclusion into AMS practices was the introduction of the 7-valent and 13-valent pneumococcal conjugate vaccine (PCV7/13) national immunisation program in South Africa [25]. Surveillance data analyses showed that the rates of disease caused by non-susceptible pneumococcal isolates were reduced by 57% in adults following the introduction of PCV7/13 national immunisations in South Africa [49]. Furthermore, a 34% and 14% reduction in invasive pneumococcal disease (IPD; all serotypes) was reported in adults 25–44 and 45–64 years of age, respectively. Similarly, the World Health Organisation (WHO) Action Framework for Leveraging Vaccines to Reduce Antibiotic Use and Prevent AMR cited evidence of the impact of PCV13 on the rates of drug-resistant IPD in the US [19]. Surveillance data reported that after the introduction of PCV13 in the US, the incidence of drug-resistant IPD decreased by 12–32% in adults within 3 years [7]. However, after the introduction of the PCV7 vaccine in the US, in areas where antibiotic prescribing remained high, the proportion of non-susceptible IPD due to non-vaccine serotypes also remained high [50].

Other vaccines discussed in the context of vaccination as part of AMS practices included the typhoid conjugate vaccine (n = 5; TCV); tetanus, diphtheria, and pertussis (n = 4; Tdap/Td); Haemophilus Influenzae type B (n = 4; Hib); measles, mumps, and rubella (n = 2; MMR); herpes zoster (n = 2; HZ); varicella (n = 2, meningococcal (n = 2); COVID-19; and hepatitis A/B (n = 1) vaccines. Cited evidence to support these vaccines was limited.

#### 3.3.2. Types of AMS Strategies including Vaccines

The results of the different types of AMS strategies identified that involved vaccines are summarised in Table 4. Eight different AMS activities involving vaccines were identified. This included education to support vaccine uptake, screening vaccination status and eligibility, appropriate vaccinations, opportunistic and targeted population vaccination, intervention alerts, counselling checklists, safety monitoring, and stock control. Four common themes or areas emerged of AMS strategies involving vaccines: Education, Screening, Vaccination, and Monitoring (Figure 3).

#### 3.3.3. Education

The most discussed AMS strategy involving vaccines, recommended by both guidelines and other articles, was education on vaccines to support uptake [23,24,25,26,31,32,33,35,38,41,51,52]. An expert opinion by Nori et al. highlighted AMS professionals as trusted sources of information as they have the expertise and skills to address vaccine hesitancy, especially for newer vaccines lacking long-term safety data [38]. The USA National Vaccine Advisory Committee (NVAC) guidelines recommended that AMS educational efforts should include information on both bacterial and viral vaccines, as well as the direct and indirect mechanisms through which they prevent AMR [31]. Several expert opinion articles supported this and recommended that AMS strategies include information on appropriate antibiotic usage while also aiming to increase vaccine uptake, as the two work synergistically to reduce AMR [24,25]. A systematic review by King et al. to identify the effectiveness of educational interventions demonstrated how AMS education positively influenced vaccine uptake [35]. Two included interventional studies that evaluated AMS education in USA Latino populations reported an increase in influenza uptake with culturally appropriate home-based education [51,52]. The first study (N = 422) reported a 10.2% increase in influenza uptake among households following the educational intervention [51]. Similarly, the second study (N = 509) reported increased influenza vaccination rates (19% to 57.1%) across all education groups post-intervention [52]. The education provided was native-language information on symptoms of the common cold and influenza, prevention strategies, locations for influenza vaccination, and a USA Centers for Disease Control and Prevention (CDC) information leaflet on the appropriate use of antibiotics.

Multiple articles reported that education and support of immunisation should be provided as part of AMS practices to both adults and children across a variety of healthcare settings [23,26,33]. According to the Australian Commission on Safety and Quality in Health Care (ACSQHC) guidelines, AMS strategies should include patient and carer education to support immunisation across all healthcare settings [21]. Resources, such as videos and posters on immunisation and AMR in health professional waiting rooms, help prepare the public before a consultation. Education on immunisations should be provided in aged care homes, the community, and hospital settings, and can be supported by physicians, pharmacists, and nurses [21,38]. However, a USA cross-sectional survey by Revolinski et al. of pharmacist understanding of AMS practices (N = 123) found that assessing and recommending vaccines was not commonly performed across healthcare settings in the future [41]. When pharmacists were asked which AMS practices they were likely to implement in the future, vaccine recommendations was one of the most common responses. Similarly, results of a 4-week interventional study by Hawksworth et al. that evaluated the use of a pilot antibiotic counselling checklist in a community pharmacy setting (N = 211) reported that only 18% of patients were counselled on influenza vaccination when antibiotics were prescribed. [32]. Barriers to the use of the checklist, as described by the participating pharmacists, included “too many options” and “very busy in the pharmacy”.

#### 3.3.4. Screening

Checking patients’ vaccination status and ensuring their immunisations are up to date was recommended by the ACSQHC guidelines across all healthcare settings [21,22]. A systematic review by Wilby et al. stated that queries on patient vaccination status and eligibility should become part of AMS programs in both inpatient and outpatient settings [46]. One of the methods suggested in the reviewed literature was the use of electronic healthcare records to acquire information on immunisations and the use of alerts/prompts to trigger opportunities for clinical support and education [21]. Surveillance systems that trigger alerts for vaccine eligibility and other AMS opportunities were indeed shown to increase interventions and recommendation acceptance in a pre-/postimplementation USA study [53], as highlighted in the 2018 ACSQHC guidance report [21]. However, the ACSQHC also recognised that alert fatigue and lengthy system processes hinder the overall success of such measures. Another USA interventional pre-post study (N = 2186) by Nowak et al. demonstrated that data-mining software could be used to generate automated screening reports for vaccination status along with other AMS intervention opportunities (e.g., recommendations to switch from intravenous to oral antibiotics) in a hospital setting, and this then reduced nosocomial infections and antimicrobial expenditure by 9.75% (~USD 1.7 million) after one year of implementation [39]. The PAM Work Group of quality measures for health-system pharmacy identified influenza vaccination screening as a key AMS measure for which pharmacy departments can and should assume accountability [23]. Several expert opinions and editorials included in this review echoed this sentiment and highlighted that to reduce AMR, in addition to ensuring the judicious uses of antimicrobials, pharmacists should check with all of their patients to ensure they are up to date with all of their immunisations [28,30,36]. Gallagher et al. further asserted that assessments on vaccination status can be performed when dispensing any medication, even non-antibiotic-related prescriptions [28]. Screening vaccination status and eligibility was also highlighted by Jorgoni et al. as a critical role of nurses to combat AMR [34].

#### 3.3.5. Vaccination

Administration of vaccines was highlighted as a key component in the reduction of AMR and a crucial component of a comprehensive AMS strategy [24,25]. A systematic review by Wilby et al. concluded that, based on evidence from both randomised control trials and epidemiological studies showing that immunisation programs reduce antimicrobial use (as discussed earlier in this review), immunisations should be considered as part of AMS programs [46]. Another systematic review by Doherty et al. also found that the existing data support that healthcare systems should prioritise the uptake of influenza and pneumococcal vaccines as part of AMS practices [27]. Where regulations permit, pharmacists should be trained and certified in administering appropriate immunisations, as well as being up to date with their own vaccinations [36]. Several expert opinions supported the role of pharmacists across various healthcare settings in the administration of vaccines as part of AMS practices [28,30,42]. Vaccinations in community pharmacies improve patient access to immunisation programs and reduce the risk of certain infections, which consequently reduces antimicrobial use and AMR [42].

Regarding targeted populations, according to multiple articles, settings where people are more vulnerable to infections (e.g., aged-cared homes) and patients at high risk of poor outcomes (e.g., the elderly and patients with chronic obstructive pulmonary disease) were recommended to be prioritised for AMS immunisation programs [29,44,45]. A report of the AMS activities from 2018–2019 highlighted that the complete staff from 172 hospitals in South Australia had been vaccinated for influenza and that offering vaccination at discharge to patients in hospitals was being considered [43]. Similarly, a narrative review by Garau et al. suggested that elderly patients who are hospitalised for respiratory tract infections be vaccinated for influenza or pneumococcal upon being discharged [29]. Brink et al. highlighted that post-partum mothers can also be prioritised for appropriate vaccinations in order to reduce illness, healthcare visits, and antibiotic use among infants [24]. Furthermore, the ACSQHC guidelines recommended that patients should be made aware that they should see their GP to receive any recommended vaccines before travelling overseas [21]. Finally, the National Health and Medical Research Council Australian Guidelines for the Prevention and Control of Infection in Healthcare recommend that AMS interventions, including immunisation, should be focused on areas of high antimicrobial use as they have been shown to generate high rates of AMR (e.g., intensive care, haematology, oncology, and transplant units) [37].

#### 3.3.6. Monitoring

Several of the reviewed articles highlighted the need for ongoing monitoring and that AMS pharmacists and physicians may be best placed to ensure the appropriate preparation and use of vaccines, as well as to monitor their safety [36,38]. Nori et al. discussed the need to expand AMS programs to include COVID-19 vaccines and emphasised the role of antimicrobial stewards in supporting the appropriate preparation and allocation of COVID-19 vaccines to healthcare workers and patients [38]. Furthermore, given the lack of long-term safety data with COVID-19 vaccines, the monitoring and reporting of the side effects of vaccines was suggested as an important task that could form part of AMS responsibilities. The authors suggested that this would complement existing allocation systems and preauthorisation paradigms for COVID-19 therapeutics. Within the pharmacy setting, an editorial by Lee et al. outlined the importance of monitoring vaccine inventory, including ensuring adequate stock for seasonal vaccines and the tracking of batch numbers and expiry dates to ensure the appropriate utilisation of vaccines [36].

## 4. Discussion

### 4.1. Summary of the Evidence to Support Vaccines as Part of AMS Practices

To the best of our knowledge, this is the first systematic scoping review evaluating the role of vaccines as part of AMS practices. The results demonstrate that the concept of incorporating vaccines as part of AMS practices is recognised, primarily based on evidence supporting the direct and indirect effects of vaccines on AMR. While this evidence is largely based on expert opinions, there were several guidelines and recommendations, systematic reviews, interventional studies, and observational studies providing higher levels of evidence supporting the inclusion of vaccines into AMS strategies. Among vaccines, influenza had the most evidence for inclusion, followed by the pneumococcal.

The literature included in this review highlight that influenza vaccinations reduce AMR by preventing influenza infections and reducing the inappropriate use of antimicrobials for respiratory symptoms, secondary bacterial infections, and hospitalisation [17,19]. This review found that influenza vaccination can reduce antimicrobial use by up to 64% among adults and likely reduces the selection pressure that drives AMR [15,18,27]. Although the evidence for the indirect impact of influenza vaccination on drug-resistant bacteria (e.g., methicillin-resistant S *aureus* [MRSA]) was not reported in the included articles, a recent study showed a clear seasonality to AMR that follows the trends in influenza rates [54]. A retrospective study using a large USA nationwide database demonstrated that the antibiotic-resistant *Streptococcus pneumoniae* and respiratory MRSA rates of hospital admissions were significantly associated with influenza rates [54]. Moreover, influenza vaccination as part of AMS is a NQF-endorsed measure, which is estimated to save the USA healthcare system billions of dollars each year. As such, the results of this review present strong evidence to support the inclusion of annual influenza vaccination as part of AMS practices [15,23].

Another vaccine of focus in the reviewed literature was pneumococcal. Evidence from this review highlight that this vaccine directly reduces the need for antibiotics, and reduces the carriage, transmission, and prevalence of drug-resistant IPD by up to one-third of baseline values. A further reduction in the overall antibiotic usage is also achieved indirectly with high vaccine coverage through herd immunity, which may further contribute to the vaccine’s ability to reduce AMR [13,24]. When looking at whether pneumococcal vaccination reduced overall antibiotic usage, the positive evidence was limited to children and was outside of the target population of this review. Interestingly, a recent randomised controlled study of adults ≥65 years of age failed to find a significant reduction in antimicrobial use in primary care with PCV13 vaccination [55]. This reinforces the fact the vaccination alone is not enough to overcome habitual antimicrobial prescribing practices and stresses the need for concurrent, judicious antibiotic utilisation intervention as part of AMS practices. With the availability of new higher-valent pneumococcal conjugate vaccines, the effects on AMR may be improved and warrant further investigation.

When looking at the types of vaccine AMS interventions, education to support vaccine uptake emerged as the most discussed and recommended AMS strategy across all healthcare settings. Increased health literacy among adults is generally associated with a more positive attitude towards vaccination and higher vaccine uptake rates [56]. The results of this review showed that AMS education on vaccines can increase influenza vaccine uptake by 10–57% among the general population, supporting the value of vaccine-related persuasive AMS interventions [35]. Details on the exact types of education materials and topics were often limited, but the key considerations identified for vaccine AMS educational interventions include coupling information on the appropriate usage of antibiotics with education on vaccines, including how vaccines reduce AMR. This can be achieved via the use of resources such as patient materials, videos, and posters, based on trusted sources of information (e.g., government sources), tailored to the target audience (e.g., native language, age, healthcare setting), and delivered at points-of-vaccination and/or opportunistically (e.g., clinic waiting-rooms, dispensing medications). Counselling tools, such as checklists, may also be used; however, they should be simple for stewards to complete and implement.

Screening of vaccination status and eligibility was also recommended as part of routine AMS practice across all healthcare settings. This can be conducted both in consultation with patients and through the screening of healthcare records, providing further opportunities for clinical support and AMS education. Despite vaccine availability and access through national immunisation programs, low vaccination rates among adults are often reported [57]. One frequently cited factor is missed opportunities for vaccination (MOV), which refers to any contact with health services by an individual (an adult or child) who is eligible for a vaccine but does not result in vaccination [58,59]. In a primary care setting, MOV of adults are reported to be as high as 95% of patient visits [58]. As such, reducing MOV is a key strategy, supported by the WHO, to increase immunisation coverage by making better use of the existing points of vaccination throughout the healthcare system [59]. The screening of vaccination status and eligibility of adults as part of AMS practices would likely help to reduce MOV, increase vaccine coverage, and consequently reduce antimicrobial use and AMR. One way to implement this into practice is to automate the process using existing surveillance systems that trigger alerts for vaccine eligibility and other AMS intervention opportunities. The results of this review found that this approach could reduce healthcare-associated infections and antimicrobial spending by approximately 10% in just one year [39]. However, care should be taken on the number of alerts triggered to avoid fatigue and to maximise intervention success. Regardless of automated systems, the screening of vaccination status and eligibility of patients can be performed by all stewards, including nurses and pharmacists, in addition to ensuring the judicious use of antimicrobials.

In this review, vaccination was found to be a crucial component of a holistic AMS strategy. As part of AMS practices, uptake prioritisation of particular vaccines, for example influenza and pneumococcal, should be considered given the available evidence discussed earlier. A recent review by van Heuvel et al. found that influenza and pneumococcal were the most frequently discussed vaccines in AMR national action plans globally [60]. However, less than half of the countries included (49%; n = 33/66) had specific measures to promote vaccination, highlighting the potential value of AMS vaccine interventions. In addition to prioritising the uptake of particular vaccines among adult patients, the uptake of vaccines among healthcare professionals (HCPs) should be supported. A survey conducted by the WHO in 2014–2015 reported low influenza vaccine uptake (median of 29.5%) among HCPs in 26 European countries [61]. A silver lining of the COVID-19 pandemic is the increased acceptance and uptake of influenza vaccinations among healthcare workers and the general public [62,63] However, there often remains a discrepancy between healthcare settings, with aged and long-term care homes reporting lower vaccine coverages despite providing care for high-risk populations [64]. These findings are consistent with the results of this review, which found that prioritising vaccinations in populations and settings where patients are more susceptible to infections and at high risk of poor health outcomes as a potential AMS strategy. As the immune system declines naturally with age, due to immunosenescence, older adults are at an increased risk of infections and age-related conditions (e.g., diabetes, cardiovascular disease) [65]. This is important given that the population is rapidly ageing and the prevalence of chronic conditions is increasing in first-world countries. The elderly also represent a reservoir for infectious diseases and can present a significant burden on healthcare systems [66]. Vaccination strategies aimed at adults and the elderly have been shown to significantly reduce morbidity and mortality due to the reduction in vaccine-preventable diseases [66]. Therefore, improvements in vaccinations among adults and high-risk populations are both a key AMS and public health strategy, and are considered one of the most impactful and cost-effective measures. Similar to childhood immunisation, innovative health policies and funding for adult immunisation as part of AMS are needed in order to make it the standard of care for high-risk populations.

Another way to improve vaccinations as part of an AMS strategy is to increase patient access to vaccines. This can be achieved by expanding the types of vaccinators and points of vaccination, for example pharmacists and pharmacy-based vaccinations. A 2020 International Pharmaceutical Federation (FIP) report found that pharmacists’ role in immunisation was expanding, with 26 countries permitting pharmacy-based vaccination at the time [67]. The COVID-19 pandemic has helped to accelerate the expansion of the role of pharmacists as vaccinators by highlighting the importance of patient access to care [68]. This, coupled with opportunistic vaccinations—for example, during patient discharge from hospital, post-partum check-ups, and/or prior to patient travel—may further improve AMS interventions targeted at vaccinations.

Other potentially beneficial AMS intervention opportunities include monitoring for reactogenicity and safety after vaccination, following up with patients to ensure series completion, and ensuring the quality use of vaccines, including appropriate storage, preparation, and administration. Where vaccine development previously took 15 years or longer, the COVID-19 pandemic has accelerated regulatory pathways, with new vaccines now able to enter the market in just 10–18 months [69]. This is significant as in surveys of self-reported ‘vaccine hesitant’ individuals, a lack of long-term data is provided as one of the main reasons for their beliefs [70]. In the absence of long-term data, post-marketing safety monitoring is crucial to better understand the benefit-risk profiles of new vaccines and can feedback into AMS educational interventions to help address vaccine hesitancy. Furthermore, many adult vaccines require multiple doses for optimal efficacy. Patient reminder systems have been shown to improve vaccine series completion and overall immunisation rates [71,72]. The implementation of patient follow-up protocols as part of AMS practices may further improve vaccine completion, consequently reducing antimicrobial usage and AMR.

Several barriers hinder the successful integration of vaccines into AMS practices, including low health literacy, vaccine hesitancy, and time constraints [32,38]. Misinformation on vaccines is highly prevalent and exposure to this has been linked to vaccine hesitancy among the general population and healthcare workers [73,74]. Education is often cited as a key AMS strategy; however, there is limited information on how to develop and deliver vaccine education to patients, the public, and/or healthcare workers in a way that instils confidence and supports uptake. A systematic review and meta-analysis by Scalia et al. found that shared decision-making interventions can be used to help address vaccine hesitancy, and increased vaccine uptake by 45% compared to the control groups [75]. AMS vaccine education delivered in this way may potentially improve the overall impact of these types of initiatives. Furthermore, education supported by tailored patient materials and/or decision aids distributed through both traditional and digital channels may also help to overcome time constraints by stewards.

### 4.2. Limitations and Future Directions

Although there was significant evidence for the impact of vaccines on AMR and a rationale for their inclusion as part of AMS practices, there was limited evidence on the impact of the implementation of vaccines as part of AMS practices. Furthermore, most of the evidence was for influenza and pneumococcal vaccines. However, there are other potential vaccines for which AMS interventions to support uptake may reduce antimicrobial usage and, consequently, the development on AMR. For example, with the emergence of new SARS-CoV-2 variants, COVID-19 vaccines are likely to become seasonal. Vaccine-related AMS interventions to support COVID-19 vaccine coverage may reduce inappropriate antimicrobial use and AMR caused by inappropriate antimicrobial use. Even with the viral origin of COVID-19, a 2020 review of cases from China and the USA reported that more than 70% of patients received antimicrobial treatments, even though less than 10% of patients reported having bacterial or fungal coinfections [76]. Similarly, herpes zoster is a vaccine-preventable disease that is estimated to affect around one in three adults during their lifetime [77]. Although also a viral infection, up to 30% of patients with herpes zoster suffer from secondary bacterial infections, for which antimicrobials are often prescribed. Supporting the uptake of other recommended adult vaccines beyond influenza and pneumococcal may provide further opportunities for stewards to reduce AMR.

Moreover, limited evidence was found on vaccine AMS interventions in immunocompromised patients, despite being considered a high-risk population to target and a source of high antimicrobial use and AMR. While targeted vaccination guidelines are available, vaccination coverage among immunocompromised populations is often low [78,79]. Vaccine AMS interventions to increase uptake in these populations may have a greater impact on reducing AMR and should be explored in future studies. In line with the WHO Action Plan [80], further research on the impact of adult vaccines on AMR and their implementation into AMS practices is warranted.

Finally, this scoping review focused on breadth rather than depth of information, precluding a meta-analysis. Similarly, the study design meant that the risk of bias of the included research articles could not be assessed. Additionally, only studies in English were included, which may limit the generalisability to English-language healthcare systems. Nevertheless, for our objective of providing an overview of the literature supporting the role of vaccines as part of AMS practices, the scoping review approach was appropriate and effective.

## 5. Conclusions

Overall, the results of the review demonstrate that the current body of evidence supports the role of vaccines as part of AMS practices. The review has scoped and identified a range of vaccine-related AMS interventions that have been utilised and/or recommended, which can be classified into four key themes (Education, Screening, Vaccination, Monitoring). Along with the judicious use of antimicrobials, implementing vaccines as part of a comprehensive AMS strategy is likely to synergistically reduce antimicrobial use and, consequently, AMR.

## Figures and Tables

**Figure 1 antibiotics-12-01429-f001:**
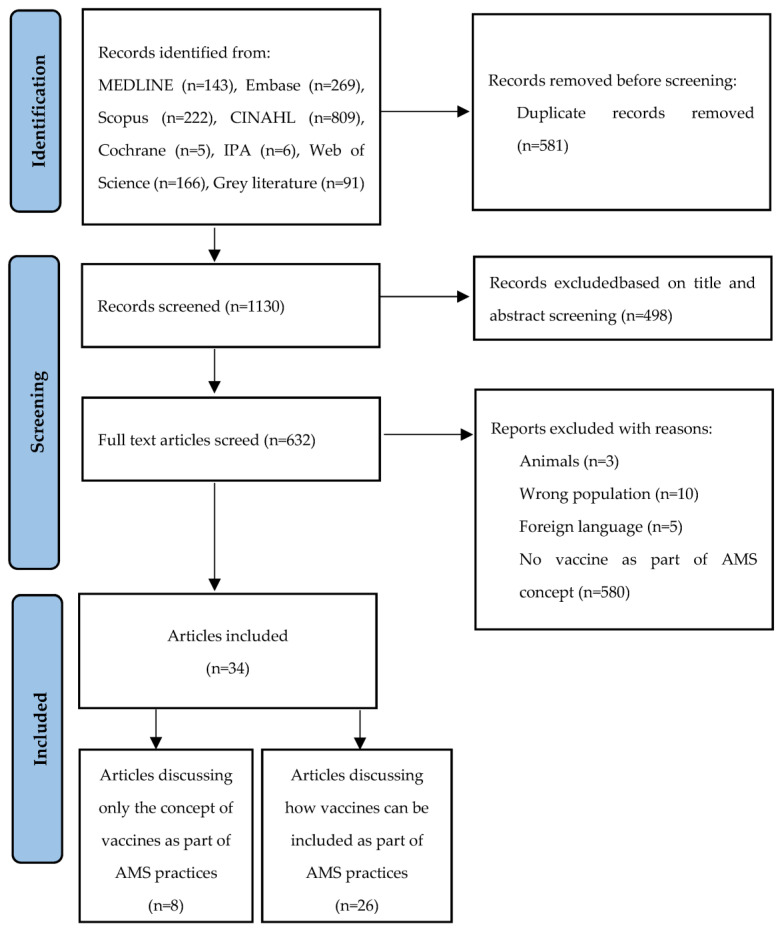
Flow chart for retrieval of articles.

**Figure 2 antibiotics-12-01429-f002:**
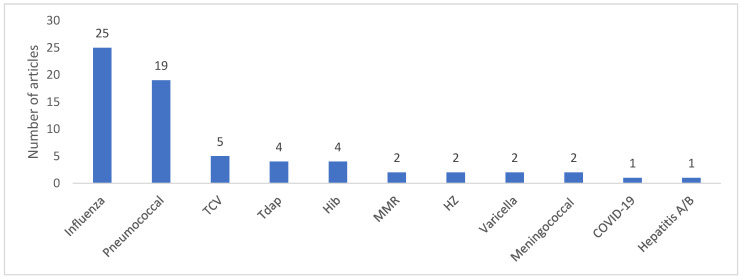
Vaccines discussed for inclusion as part of AMS practices. HZ, herpes zoster; MMR, measles, mumps, and rubella; TCV, typhoid conjugate vaccine; Tdap, tetanus, diphtheria, and acellular pertussis.

**Figure 3 antibiotics-12-01429-f003:**
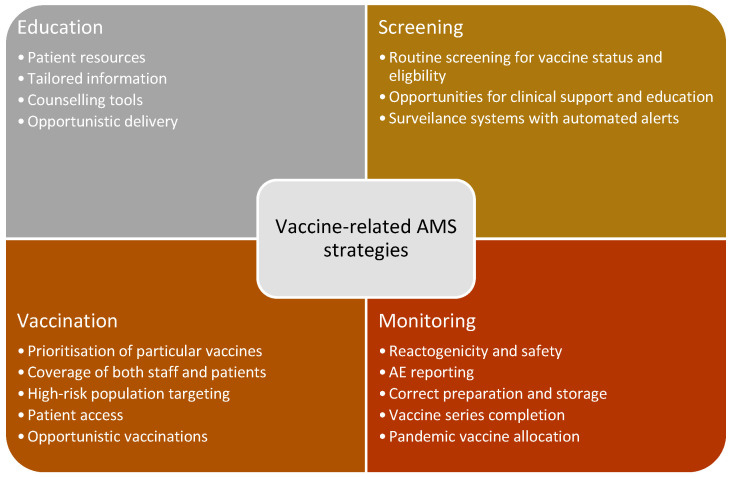
Identified key themes of vaccine-related AMS strategies.

**Table 1 antibiotics-12-01429-t001:** Inclusion and exclusion criteria.

Inclusion	Exclusion
Adult patients or individuals (≥18 years of age)	Adolescent and paediatric patients or individuals (<18 years of age)
Vaccinations against both bacterial and viral infections	Animal studies
All sources of evidence (primary studies [RCTs, cohort studies, case-control, cross-sectional studies, case series or reports], systematic reviews, reports, dissertations, expert opinion articles, grey literature [.org, .edu, .int, .gov, .au])	Articles in any language other than English
English language articles from all geographical locations	

**Table 2 antibiotics-12-01429-t002:** Research articles discussing only the concept of vaccines as part of AMS practices.

Study Author	Year of Publication	Country	Evidence Type	Objective	Population/Design	Key Findings	Vaccine Example/Target	Limitations/Gaps
Buchy et al. [13]	2020	Singapore	Expert opinion	To review how vaccines may be used to combat against AMR.	Narrative review on the impact of vaccines on AMR across healthcare settings.	Combining vaccine and non-vaccine approaches is valuable for antimicrobial stewardship (AMS) strategy. Large-scale vaccination programs are more feasible to implement than antibiotic restriction policies. Different vaccination programs have shown evidence of reducing antibiotic-resistant infection, morbidity, and mortality rates.	Pneumococcal, influenza, Tdap, Hib, TCV	No information on how vaccines should be included as part of an AMS strategy.
Curcio et al. [14]	2017	Argentina	Retrospective observational study	To identify the antibiotics prescription patterns for adults hospitalised with CAP.	Retrospective study (N = 1697) using a health insurance database of adults (≥18 years old) treated for antibiotics (≥3 days) for CAP.	Patients ≥65 years old were significantly associated with CAP multi-morbidity, ≥7-day antibiotic use and broad-spectrum use, and intensive care unit admission. The authors state that prevention strategies for CAP, such us pneumococcal and influenza vaccinations, must be considered as part of the AMS strategy.	Pneumococcal, influenza	Abstract only. Study did not assess the impact of vaccination status on prescription patterns. No information on how vaccines should be included as part of an AMS strategy.
Darazam et al. [15]	2019	Iran	Expert opinion	To review the evidence of the direct and indirect effects of influenza and vaccination on health systems.	Narrative review on the impact of influenza and vaccination on AMR in both a hospital and community setting.	Reduced antibiotic prescriptions was reported with the introduction of universal influenza immunisation and immunisation of post-partum mothers. Authors declared this as definitive evidence to consider annual influenza vaccination in AMS programs.	Influenza	No information on how vaccines should be included as part of an AMS strategy.
Feldman et al. [16]	2018	South Africa	Expert opinion	To review the appropriate antibiotic management of bacterial LRTIs.	Narrative review of antibiotic use mainly in AECOPD and CAP in both a hospital and community setting.	AMS initiatives are needed to manage AECOPD and CAP, aiming to slow AMR rates and improve patient outcomes. Vaccination aligns with AMS principles and can effectively reduce antibiotic usage. A meta-analysis found that oral vaccination against NTHi in AECOPD patients led to an 80% reduction in antibiotic courses compared to non-vaccination.	Pneumococcal, influenza, Hib	No information on how vaccines should be included as part of an AMS strategy. Difficulty generalising results to overall population due to specific patient population studied.
O’Brien et al. [17]	2013	UK	Expert opinion	To review innovative mechanisms to complement existing AMS initiatives	Narrative review of AMS initiatives in a general public, community, and hospital setting.	Vaccines reduce disease prevalence and the need for antibiotics, supporting their role in AMS. New vaccine development and immunisation programs indirectly contribute to AMS.	Pneumococcal, influenza	No information on how vaccines should be included as part of an AMS strategy.
Smith et al. [18]	2019	USA	Prospective observational study	To evaluate the proportion of antibiotic prescriptions in the community with ARI that can be avoided by influenza vaccination.	Patients aged ≥6 months with ARI (N = 37,847) were followed from 2013–2014 through 2017–2018 influenza seasons at >50 community clinics. and the proportion of ARI antibiotic prescriptions averted by influenza vaccination was estimated.	Influenza vaccinations improved antibiotic prescribing in a community setting. Influenza vaccination was prevented 10.6% of ARI syndromes and averted 7.3% antibiotic prescriptions in patients with ARI. Stewardship through influenza vaccination coverage may help achieve national goals of reducing AMR.	Influenza	Abstract only. No mean age reported or stratification of results by age, making the impact in adults vs. children hard to discern. No information on how vaccines should be included as part of an AMS strategy.
WHO/Vekemans J et al. [19]	2021	Switzerland	Guidelines/recommendations	To guide vaccine stakeholders’ efforts in maximising the impact of vaccines in preventing and containing AMR.	Consensus-based recommendations on leveraging vaccines to reduce AMR across healthcare settings.	The 2030 WHO immunization agenda aims to expand knowledge on the impact of vaccines on AMR. Researchers should generate new evidence on how vaccines complement AMS. Single and combination vaccines have synergistic effects on antimicrobial use and AMR, reinforcing the role of vaccines as an AMS tool.	Pneumococcal, TCV, influenza, MMR, Hib.	No information on how vaccines should be included as part of an AMS strategy.
Wuethrich et al. [20]	2021	Germany	Expert opinion	To review available evidence on non-traditional interventions that reduce AMR and their interaction with the human microbiota.	Narrative review on interventions that can reduce AMR, support AMS and the gut microbiota in a hospital setting.	Vaccines reduce the use of antibiotics, which reduces the selection pressure on AMR and aligns with AMS measures. Vaccines also reduce bystander selection of resistant strains in the gut microbiota. Vaccines potentially enhance the overall effectiveness of AMS programs.	None provided	No information on how vaccines should be included as part of an AMS strategy.

ACSQHC, Australian Commission on Safety and Quality in Health Care; AECOPD, acute exacerbations in chronic; AMR, antimicrobial resistance; AMS, antimicrobial stewardship; ARTI, acute respiratory tract infection; CAP, community acquired pneumonia; Hib, *H. influenzae* type B; ICU, intensive care unit; LRTI, lower respiratory tract infections; NTHi, non-typeable *H. influenzae* type B; PACCARB, Presidential Advisory Council on Combating Antibiotic-Resistant Bacteria; RCT, randomised control trial; TCV, typhoid conjugate vaccine; URTI, upper respiratory tract infection; UTI, urinary tract infections; WHO, World Health Organization.

**Table 3 antibiotics-12-01429-t003:** Research articles discussing the concept of vaccines as part of AMS practices and vaccine-related AMS interventions.

Study Author	Year of Publication	Country	Evidence Type	Objective	Population/Design	Key Findings	Vaccine Example/Target	Limitations/Gaps
ACSQHC [21]	2018	Australia	Guidelines/ recommendations	To outline the Australian national framework for AMS.	Evidence-based recommendations on key elements of effective AMS programs across healthcare settings.	AMS strategies require education across all healthcare settings (aged care homes, community, hospital settings) for patients and carers, supported by pharmacists and nurses. Travellers should be educated to receive recommended vaccines before travelling. Use of EHR systems with trigger alerts for vaccine eligibility/other AMS opportunities can increase interventions and vaccine acceptance. However, alert fatigue and lengthy system processes hinder overall success.	Pneumococcal, influenza	Limited information on how to educate patients and carers to support immunisation.
ACSQHC [22]	2021	Australia	Guidelines/ recommendations	To outline AMS practices in community and residential aged care.	Evidence-based recommendations to improve AMS in community and aged-care settings.	Widespread antimicrobial use for URTIs/acute bronchitis without bacterial confirmation is an area of potential misuse. AMS opportunities include checking vaccination status and ensuring immunisations are up to date.	None provided	No information on which vaccinations to check or prioritise. No information on how to educate and encourage vaccinations.
Andrawis et al. [23]	2019	USA	Systematic review	To identify recommended accountability measures for pharmacy departments according to the PAM Work Group.	Systematic review of National Quality Measure Clearinghouse and NFQ-endorsed measures (N = 656 measures) for pharmacy across healthcare settings.	The PAM Work Group identified 28 measures. They emphasised the role of pharmacists in screening, recommending, and conducting immunisations as part of AMS in hospitals and communities. Influenza screening and vaccination was a key endorsed measure.	Influenza	Database search methodology not defined with room for further optimisation. Other vaccinations beyond influenza were not assessed with no rationale provided.
Brink AJ [24]	2016	South Africa	Expert opinion	To review components of AMS programs for ARTIs in the community and propose new AMS strategies.	Narrative review of AMS practices in a community setting.	Vaccination is crucial in reducing AMR and an essential part of AMS strategy. Effective AMS campaigns should include multifaceted educational interventions, including vaccination, targeting community-specific barriers.	Pneumococcal, influenza, Tdap, Hib	Limited information on how to educate and implement a campaign that targets both AMR and vaccine uptake. No information on what the expected barriers to uptake are and how to overcome them.
Brink et al. [25]	2015	South Africa	Expert opinion	To review vaccination and AMS programs.	Narrative review of vaccines and their impact on AMR and antibiotic usage.	Vaccines are a key AMS strategy. PCV13 reduces drug-resistant pneumococcal disease by impacting nasopharyngeal carriage, otitis media, and pneumonia. It also reduces carriage and disease in vaccine non-eligible individuals. Influenza vaccination prioritisation, especially for postpartum mothers, is beneficial.	Pneumococcal, influenza, Tdap, Hib	Limited information on how vaccines should be included as part of an AMS strategy and how to educate to increase uptake.
Caffery et al. [26]	2019	USA	Editorial	To highlight the role of nurses in AMS.	Discussion of the role of nurses in a hospital setting.	The role of nurses in AMS is to ensure the safe and appropriate use of antimicrobials and includes the recommending immunisations for adults.	None provided	Non-peer reviewed. No information on how immunisation should be recommended as part of AMS practices.
Doherty et al. [27]	2020	Belgium	Systematic review	Review current data on the impact of vaccination on antimicrobial usage.	Systematic review of RCTs, observational, and population-based database influenza and pneumococcal studies (N = 26). All patient populations across healthcare settings were included. Narrative summary was then conducted.	In adults, influenza vaccination led to a 52% reduction in antimicrobial usage. Pneumococcal vaccination decreased overall antimicrobial prescribing by 2–19%. Healthcare systems should educate and prioritise the uptake of these vaccines as part of AMS practices.	Pneumococcal, influenza	No meta-analysis performed. Majority of pneumococcal data were from children; difficult to discern the impact in adults vs. children. Limited information on how to educate and support uptake.
Gallagher et al. [28]	2017	USA	Expert opinion	The review aims to cover AMS mandates, teaching approaches for required skills, and best practices in training pharmacists as antimicrobial stewards.	Discussion on the training of future pharmacists as antimicrobial stewards in a community setting.	In community pharmacies, an AMS strategy component involves screening, recommending, and administering influenza and pneumococcal vaccines as needed. Assessing vaccination status can be conducted even without antibiotic prescriptions.	Pneumococcal, influenza	Limited to the training of pharmacists. No direct evidence for the impact of vaccines on AMR or rationale for focus on particular vaccines versus others provided.
Garau et al. [29]	2014	Spain	Expert opinion	To review AMS challenges and strategies to prevent rising AMR.	Narrative review of AMS practices in an hospital and community setting to manage RTIs and UTIs.	AMS for RTIs should include immunisation programs for high-risk individuals, such as COPD patients. Vaccination against bacterial infections reduces community-acquired infections and improves antibiotic use. Elderly patients hospitalised for RTIs should receive pneumococcal or influenza vaccination upon hospital discharge. Immunisation strategies for UTIs, although undeveloped, have the potential to reduce antibiotic use and AMR.	Pneumococcal, influenza, meningococcal	Limited information on patient discharge eligibility requirements for vaccination.
Gershman J [30]	2019	USA	Editorial	To highlight how pharmacists play an important role in AMS strategies.	Discuss pharmacists’ roles in AMS programs in a community and hospital setting.	To reduce AMR, in addition to judicious use of antimicrobials, pharmacists should check with patients to ensure they are up to date with all immunisations. Pharmacists can then administer any needed vaccines.	None provided	Non-peer reviewed. Limited information on when and how to screen/recommend vaccinations.
Gordon et al. [31]	2015	USA	Guidelines/ recommendations	Outline NVAC recommendations for combatting AMR.	Evidence-based recommendations on use of vaccines to combat AMR across healthcare settings.	The NVAC recommends including education on the role of vaccines in reducing antibiotic use within AMS stakeholder efforts. New vaccine development may also combat AMR.	Pneumococcal, influenza, Hib	Limited information on how to increase uptake as part of AMS practices and when it is best to communicate the role of vaccines in reducing antibiotic use.
Hawksworth et al. [32]	2019	UK	Interventional study	To evaluate use of a pilot RPS antibiotic checklist in community pharmacies.	The RPS checklist along with a patient information leaflet was used for 4 weeks to counsel antibiotic prescription patients in community pharmacies (N = 211 patients counselled). A self-assessment recorded number of used counselling points and perceived usefulness.	Five pharmacists found the RPS checklist useful and 3 thought it was time consuming. As part of the checklist, 18% (n = 39) of patients were advised on influenza vaccination. Barriers to use of the checklist included “too many options” and “very busy in the pharmacy”.	Influenza	Abstract only. Small study size; only a limited number of pharmacists used the checklist. No information on how immunisation was advised to patients as part of AMS practices.
Hurst et al. [33]	2013	USA	Expert opinion	To review how AMS can be applied to the management of CAP.	Narrative review of AMS initiatives for CAP in an hospital setting.	System-wide vaccination recommendations optimises CAP management and improves patient outcomes. Adult pneumococcal vaccination lowers invasive disease risk, improves survival, and shortens hospital stays. It reduces antibiotic use and combats AMR.	Pneumococcal	Limited information on how immunisation should be recommended as part of AMS practices.
Jorgoni et al. [34]	2017	Canada	Editorial	Q&A on the relationship between vaccines and AMR	Discussion on the role of nurses in reducing AMR in a community and hospital setting.	Patient education on the risks of antibiotic misuse and ensuring immunisations are up to date are crucial nursing roles in the battle against AMR.	Pneumococcal, influenza	Non-peer reviewed. No information on when and how to screen/recommend vaccinations.
King et al. [35]	2015	UK	Systematic review	To identify effective AMS education interventions for public behaviour change on antimicrobial use and infection prevention, reducing AMR.	Systematic review in line with NICE guidelines (N = 60) on educational interventions in a public setting with a descriptive analysis.	Two interventional studies evaluating AMS education in Latino populations reported an increase in influenza uptake with culturally appropriate home-based education.	Influenza	Limited information on what vaccination education was provided. Two relevant studies were in Latino populations which may not be generalisable to the general population.
Lee et al. [36]	2014	USA	Editorial	To explore how community pharmacists can implement AMS programs.	Discussion of AMS programs and principles for pharmacists in a community setting.	Pharmacists must prioritise and recommend immunisations. They should be certified to administer them, keep their vaccinations updated, and stock the correct inventory. Community-wide vaccinations boost herd immunity, reducing doctor visits and unnecessary antibiotic prescriptions for viral illnesses.	Pneumococcal, influenza, hepatitis A/B, varicella, meningococcal, MMR, herpes zoster, Tdap	Non-peer reviewed. No information on how and when immunisation should be recommended. No direct evidence for the impact of vaccines on AMR provided.
NHMRC [37]	2019	Australia	Guidelines/ recommendations	To outline the National approach to IPC.	Evidence-based IPC recommendations across healthcare settings.	Infection prevention is an essential part of AMS and reduces AMR. Vaccination helps by preventing infections, including viral, often mistreated with antimicrobials. Specialised units like ICU, haematology, oncology, and transplant units exhibit higher antimicrobial use and AMR rates, warranting targeted AMS interventions.	None provided	Limited information on how to implement vaccinations as part of AMS interventions and how to implement in acute and specialised settings. Predominantly drawn from the acute setting.
Nori et al. [38]	2021	USA	Expert opinion	To discuss the allocation of COVID-19 vaccines to healthcare workers and patients as part of AMS programs.	A single PubMed search was performed on COVID-19 vaccines and AMS programs in a hospital setting.	AMS pharmacists and physicians play key roles in vaccine education, preparation, monitoring, and side-effect reporting. As trusted sources, they address hesitancy, especially with new vaccines lacking long-term safety data. Expanding their responsibilities complements existing allocation systems and preauthorisation protocols for COVID therapeutics.	COVID-19	No direct evidence for the impact of COVID-19 vaccines on AMR provided. Unclear if peer-reviewed.
Nowak et al. [39]	2012	USA	Interventional study	To evaluate the clinical and economical outcomes of a hospital AMS program that utilised automated reports from data-mining software.	Quasiexperimental study design analysed hospital patient charts (N = 2186) before and after AMS program implementation. Primary outcomes were annual antimicrobial expenditures and rates of infections due to common nosocomial pathogens.	Using data-mining software, automated screening reports for vaccination status and other AMS opportunities were generated (e.g., recommendations for empirical therapy, IV to oral antibiotic switch, de-escalation, and discontinuation). In the first year of AMS program implementation, antimicrobial spending fell by 9.75% (~USD 1.7 million), and nosocomial infections decreased, including *C. difficile* and VRE.	Pneumococcal, influenza	Non-randomised. The individual effect of vaccines cannot be discerned from the other AMS interventions.
PACCARB [40]	2017	USA	Guidelines/ recommendations	To outline a framework for incentivising the development of vaccines, diagnostics, and antimicrobials to combat AMR.	Recommendations based on the outputs from three working groups consisting of council members and subject matter experts in both human and animal domains.	Health providers lack knowledge about the role of vaccines in preventing AMR. Effective use of influenza, varicella, pneumococcal vaccines in adults can significantly reduce antibiotic use. Vaccination should be included as part of AMS programs, and research and education should be conducted to improve uptake.	Pneumococcal, influenza, varicella	Limited information on how vaccines should be included as part of an AMS strategy and what type of education is needed to improve uptake.
Revolinski et al. [41]	2020	USA	Cross-sectional study	To understand pharmacy student and preceptor understanding and application of AMS practices	Pharmacy students from one university filled AMS checklists and reflections in various practice settings (inpatient, ambulatory clinic, community). A survey then analysed the AMS education impact on students (n = 60) and preceptors (n = 63).	Assessing and recommending vaccinations, 1 of 16 AMS practices, was part of the checklist and improved pharmacy students’ understanding of vaccines in AMS. Preceptors expressed interest in implementing this practice, which involved analysing patient data to determine eligibility, recommending immunisations, and administering vaccines.	None provided	No direct evidence for specific vaccines provided. Limited information on when and how to screen/recommend vaccinations.
Rosenberg-Yunger et al. [42]	2019	Canada	Expert opinion	Review of the role of community pharmacists as antimicrobial stewards.	Narrative review AMS programs and activities of pharmacists in a community setting.	Community pharmacy AMS practices should include the recommendation and administration of vaccines. Pharmacy-based immunisations improve patient access and prevent infections, reducing antimicrobial use.	None provided	No direct evidence for specific vaccines provided. No information on how/when immunisation should be promoted as part of AMS practices.
SAAGAR [43]	2020	Australia	Expert opinion	To provide an annual overview of AMS activities and achievements.	Report of AMS activities in South Australia across all healthcare sectors from 2018–2019.	172 Inpatients were vaccinated for influenza as part of a new AMS intervention of record screening for vaccine eligibility. The health network is looking to expand on this and offer vaccinations on discharge.	Influenza	Limited evidence for the impact of vaccines on AMR provided.
Thompson CA [44]	2018	USA	Editorial	To highlight the benefit of pharmacist AMS at urgent care centres.	Discussion of pharmacist AMS practices and impact in a community setting.	As part of AMS practices, pharmacists should counsel all ambulatory care and home care patients on the need to receive appropriate vaccinations.	Pneumococcal, influenza	Non-peer reviewed. Limited information on how to counsel on vaccinations.
While A [45]	2017	UK	Expert opinion	Review AMR challenges and how AMS should be implemented in care homes.	Narrative review of AMS practices by nurses in an aged-care setting.	IPC measures, such as supporting vaccine uptake are a key component of AMS, particularly in in care homes, where people are more vulnerable to infections.	Influenza, herpes zoster	Limited information on how to increase uptake of vaccinations.
Wilby et al. [46]	2012	Canada	Systematic review	To assess antimicrobial use in relation to immunisation programs or studies on vaccine effectiveness.	Systematic review of MEDLINE, EMBASE, IPA, and Google Scholar for studies (N = 7) reporting antimicrobial use in connection with vaccine use. All healthcare settings and age groups were included.	Three RCTs and four epidemiological studies were found, all showing reduced antibiotic use after introducing influenza and pneumococcal immunisation programs. RCTs saw 5–10% reductions, while epidemiological studies observed 64% reductions. Thus, vaccination status updates should be routine in both inpatient and outpatient care, and immunisation programs should be included in AMS initiatives.	Pneumococcal, influenza	Small number of articles identified. No meta-analysis performed. Difficult to discern the impact of vaccines in adults vs. children.

ACSQHC, Australian Commission on Safety and Quality in Health Care; AECOPD, acute exacerbations in chronic; AMR, antimicrobial resistance; AMS, antimicrobial stewardship; ARTI, acute respiratory tract infection; CAP, community acquired pneumonia; HER, electronic health record systems; Hib, *H. influenzae type B*; ICU, intensive care unit; LRTI, lower respiratory tract infections; NQF, National Quality Forum; NVAC, National Vaccine Advisory Committee; PACCARB, Presidential Advisory Council on Combating Antibiotic-Resistant Bacteria; PAM, Pharmacy Accountability Measures; PCV, pneumococcal conjugate vaccine; RCT, randomised control trial; RPS; Royal Pharmaceutical Society; SAAGAR, South Australian expert Advisory Group on Antimicrobial Resistance; TCV, typhoid conjugate vaccine; URTI, upper respiratory tract infection; UTI, urinary tract infections. VRE, vancomycin resistant enterococci.

**Table 4 antibiotics-12-01429-t004:** Types of AMS strategies recommended and/or utilised to support immunisations.

Research Article	Evidence Type	Type(s) of AMS Activities	Theme(s) of AMS Strategies
ACSQHC 2018 [21]	Guidelines/recommendations	Education to support vaccine uptake, appropriate vaccinations, opportunistic and targeted vaccinations, screening vaccination status and eligibility, intervention alerts	Education, Screening, Vaccination
ASCQHC 2021 [22]	Guidelines/recommendations	Screening vaccination status and eligibility	Screening
NHMRC [37]	Guidelines/recommendations	Appropriate vaccinations, opportunistic and targeted vaccinations	Vaccination
NVAC [31]	Guidelines recommendations	Education to support vaccine uptake	Education
PACCARB [40]	Guidelines/recommendations	Education to support vaccine uptake	Education
Andrawis et al. [23]	Systematic review	Education to support vaccine uptake, screening vaccination status and eligibility, appropriate vaccinations, opportunistic and targeted vaccinations	Education, Screening, Vaccination
Doherty et al. [27]	Systematic review	Education to support vaccine uptake	Education
King et al. [35]	Systematic review	Education to support vaccine uptake	Education
Wilby et al. [46]	Systematic review	Screening vaccination status and eligibility, appropriate vaccinations	Screening, Vaccination
Hawksworth et al. [32]	Interventional study	Education to support vaccine uptake counselling checklist	Education
Nowak et al. [39]	Interventional study	Screening vaccination status and eligibility, intervention alerts	Screening
Revolinski et al. [41]	Cross-sectional study	Education to support vaccine uptake, screening vaccination status and eligibility, counselling checklist, appropriate vaccinations	Education, Screening, Vaccination
Brink AJ [24]	Expert opinion	Education to support vaccine uptake, appropriate vaccinations	Education, Vaccination
Brink et al. [25]	Expert opinion	Education to support vaccine uptake, appropriate vaccinations, opportunistic and targeted vaccinations	Education, Vaccination
Gallagher et al. [28]	Expert opinion	Education to support vaccine uptake, screening vaccination status and eligibility, appropriate vaccinations	Education, Screening, Vaccination
Garau et al. [29]	Expert opinion	Appropriate vaccinations, opportunistic and targeted vaccinations	Vaccinations
Hurst et al. [33]	Expert opinion	Education to support vaccine uptake	Education
Nori et al. [38]	Expert opinion	Education to support vaccine uptake, safety monitoring.	Education, Monitoring
Rosenberg-Yunger et al. [42]	Expert opinion	Education to support vaccine uptake, appropriate vaccinations	Education, Vaccination
SAAGAR Report [43]	Expert opinion	Screening vaccination status and eligibility, appropriate vaccinations, opportunistic and targeted vaccinations	Screening, Vaccination
While A [45]	Expert opinion	Appropriate vaccinations, opportunistic and targeted vaccinations	Vaccination
Caffery et al. [26]	Editorial	Education to support vaccine uptake	Education
Gershman J [30]	Editorial	Screening vaccination status and eligibility, appropriate vaccinations	Screening, Vaccination
Jorgoni et al. [34]	Editorial	Screening vaccination status and eligibility	Screening
Lee et al. [36]	Editorial	Education to support vaccine uptake, Screening vaccination status and eligibility, appropriate vaccinations, stock control	Education, Screening, Vaccination, Monitoring
Thompson CA [44]	Editorial	Education to support vaccine uptake, appropriate vaccinations	Education, Vaccination

ACSQHC, Australian Commission on Safety and Quality in Health Care; NHMRC, National Health and Medical Research Council; NVAC, National Vaccine Advisory Committee; PACCARB, Presidential Advisory Council on Combating Antibiotic-Resistant Bacteria; SAAGAR, South Australian Expert Advisory Group on Antimicrobial Resistance.

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
