# Peer review of "The Role of Adult Vaccines as Part of Antimicrobial Stewardship: A Scoping Review"

_antibiotics, 2023, doi:10.3390/antibiotics12091429_

Round 1

Reviewer 1 Report

The choice of the theme is very well. 

The quality of the manuscript is excellent

  1. The main question is antimicrobial resistence in relation to immunisation
  2. The topic is apsolutly relevant
  3. The interaction between AMR and immunisation was rarely analised
  4. Sistematic review and meta analysis are top methods in investigation
  5. Conclusions are consistent
  6. Ref are appropiate
  7. Without comments  

Author Response

Dear Reviewer 1,

Thank you for taking the time to review the manuscript and for sharing the positive feedback.

Very much appreciated.

Kind regards,

Charles

Reviewer 2 Report

Minor edits to correct use of language is needed.

A more concise discussion can be attempted.

  The main question addressed by the research is scoping review on use of vaccines as a component of AMS strategies.   This topic is highly relevant in the current scenario of pandemic and emerging diseases.   It helps in comprehensive review of material generated.   I think the methodology adopted by the authors are appropriate for the topic and purpose.
The conclusions are consistent with the evidence and arguments presented
and address the main question posed.   Addition of more references related to the topic will be more appropriate.   The tables and figures included are sufficient for the topic discussed.

Generally good use of the language barring a few typos and incorrect usages.

Author Response

Dear Reviewer 2,

Thank you for taking the time to review the manuscript and for sharing the positive feedback.

Please find responses to your comments:

  • Minor edits to correct use of language is needed. Generally good use of the language barring a few typos and incorrect usages.

Manuscript has been reviewed and language usage/typos corrected. Thanks.

  • A more concise discussion can be attempted

As we received no other comments from the other peer reviewers on this point, we decided to keep the flow of the discussion as is. However, we took your feedback on board and copy edited the discussion down where appropriate.

  • Addition of more references related to the topic will be more appropriate. 

We have included an additional recent study on the topic by Van Heuval et al. (doi: 10.1186/s12992-022-00878-6.) to further support in the discussion section.. 

Thanks again for your inputs. 

Kind regards,

Charles

Reviewer 3 Report

This manuscript discusses the process of evaluation of the use of vaccines in prevention of antimicrobial resistance (AMR).  The manuscript is well done and this reviewer cannot ask the authors to revise the manuscript for improvement.  Well done. 

Author Response

Dear Reviewer 3,

Thank you for taking the time to review the manuscript and for sharing the positive feedback.

Very much appreciated.

Kind regards,

Charles

Reviewer 4 Report

In this papers author present a scoping review about the role of vaccines as a part of antimicrobial stewardship programs. The topic is of interest because infection prevention can be very important to reduce antimicrobial resistance, but it is not routinely regarded as a tool to be included in antimicrobial stewardship programs.  

The scoping review is appropriate for providing an overview of literature supporting the role of vaccines as part of AMS practices. The text is weel written and the overall methodology is correct. 

Some specific comments: 

Title. As the review focus on vaccination in the adult population, it should be stated in the title 

Introduction.  

Lines 57-58. The sentence Existing research indicates the lack of comprehensive information on the use of vaccines as a component of AMS strategies is not very clear and could probably be omitted  

Lines 89-90 The sentence “Furthermore, children and adolescents are often not the primary decision makes when it comes to vaccinations” seem not to be very pertinent 

The final version of the charting form should be included, even as supplementary material. 

Results 

Reason for exclusion of 498 records in the screening phase (title and abstract) is not provided in the text nor in the figure, it should be indicated. 

Table 2 and Table 3 seem to be placed in the wrong place, at the end of the text and following table 4, instead of the end of the related paragraph.

The manuscript will benefit from minor editing of English language, to improve clarity of some sentences

Author Response

Dear Reviewer 4,

Thank you for taking the time to review the manuscript and for sharing your feedback.

Please find responses to your comments:

  • As the review focus on vaccination in the adult population, it should be stated in the title

This is a valid point and has been added to the title. Thanks.

  • Lines 57-58. The sentence “Existing research indicates the lack of comprehensive information on the use of vaccines as a component of AMS strategies” is not very clear and could probably be omitted

Have omitted as suggested. Thanks.

  • Lines 89-90 The sentence “Furthermore, children and adolescents are often not the primary decision makes when it comes to vaccinations” seem not to be very pertinent.

We believe this to be an important point on the rationale to focus the review on adults vs children or adults and children. The psychology behind vaccine acceptance in adults themselves vs parents for their children differs and as such, potential AMS strategies for vaccines would likely need to be different for these two groups. Finally, as we received no other comments from the other peer reviewers on this point, we decided to keep the sentence as is. Hopefully, the rationale provided above meets your expectations. Thanks.

  • The final version of the charting form should be included, even as supplementary material. Table 2 and Table 3 seem to be placed in the wrong place, at the end of the text and following table 4, instead of the end of the related paragraph.

The final version of the charting forms (Table 2 and 3) have been placed in landscape after the first mention withing the text body. Hopefully this addresses the comment above. Thanks.

  • Reason for exclusion of 498 records in the screening phase (title and abstract) is not provided in the text nor in the figure, it should be indicated.

We have further clarified and added to figure the reason for exclusion. Thanks.

  • The manuscript will benefit from minor editing of English language, to improve clarity of some sentences.

Manuscript has been reviewed and language usage/sentence clarity corrected. Thanks.

Thanks again for your inputs. 

Kind regards,

Charles

Reviewer 5 Report

The central topic of the paper is to evaluate through a scoping review the scientific evidence regarding the role of vaccines in antimicrobial stewardship.

From a methodological point of view, the work is conducted in a rigorous manner

The methodology used for the search of the literature and the description of the results are well explained, clear and readable.

Table 2 should be moved to where it is first mentioned and should also be better organised, as it is difficult to read.

Perhaps these two recent reviews (probably published after their bibliographic analysis), but dealing with the subject, could be useful in the discussion:

van Heuvel L, Caini S, Dückers MLA, Paget J. Correction to: Assessment of the inclusion of vaccination as an intervention to reduce antimicrobial resistance in AMR national action plans: a global review. Global Health. 2023 Aug 8;19(1):54. doi: 10.1186/s12992-023-00951-8. Erratum for: Global Health. 2022 Oct 17;18(1):85. PMID: 37553582; PMCID: PMC10408207.

Micoli F, Bagnoli F, Rappuoli R, Serruto D. The role of vaccines in combatting antimicrobial resistance. Nat Rev Microbiol. 2021 May;19(5):287-302. doi: 10.1038/s41579-020-00506-3. Epub 2021 Feb 4. PMID: 33542518; PMCID: PMC7861009.

The relevance and limitations of the paper are clearly stated in the conclusions.

Author Response

Dear Reviewer,

Thank you for taking the time to review the manuscript and for sharing your valuable feedback. 

Please find the detailed responses below.

Comment 1: Table 2 should be moved to where it is first mentioned and should also be better organised, as it is difficult to read.

As per your feedback, Table 2 and 3 have been moved to first mention and placed in landscape so easier to read. Thanks.

Comment 2: Perhaps these two recent reviews [Van Heuvel et al. and Micoli et al.] (probably published after their bibliographic analysis), but dealing with the subject, could be useful in the discussion:

Good suggestion. I have referred to Van Heuval et al. in the discussion as relevant primary research in the area. We chose to exclude Micoli et al. as this was not primary research and was more focused on novel future vaccine technologies for AMR pathogens versus vaccines as part of AMS practices.  

Thanks again for your inputs. Very much appreciated.

Kind regards,

Charles